# Investigating the Longevity and Infectivity of *Cucumber green mottle mosaic virus* in Soils of the Northern Territory, Australia

**DOI:** 10.3390/plants11070883

**Published:** 2022-03-25

**Authors:** David Lovelock, Sharl Mintoff, Nadine Kurz, Merran Neilsen, Shreya Patel, Fiona Constable, Lucy Tran-Nguyen

**Affiliations:** 1Department of Jobs Precincts and Regions, Agriculture Victoria Research, Agribio, Bundoora, VIC 3083, Australia; fiona.constable@agriculture.vic.gov.au; 2Department of Industry, Tourism and Trade, Biosecurity and Animal Welfare, Darwin, NT 0801, Australia; sharl.mintoff@nt.gov.au (S.M.); nadinekurz@googlemail.com (N.K.); merran.neilsen@nt.gov.au (M.N.); shreya.patel@nt.gov.au (S.P.); 3Plant Health Australia, Level 1, 1 Phipps Close, Deakin, ACT 2600, Australia; ltran-nguyen@phau.com.au

**Keywords:** *Cucumber green mottle mosaic virus*, soil longevity, disinfection

## Abstract

*Cucumber green mottle mosaic virus* (CGMMV) is a *Tobamovirus* of economic importance affecting cucurbit crops and Asian cucurbit vegetables. CGMMV was detected in the Northern Territory (NT) in September 2014, the first record for Australia, with 26 properties confirmed as of May 2016. Research was undertaken to determine virus longevity in soils in the NT and investigate the use of disinfectants to remove viable CGMMV from the soil. An in-field trial at 12 months post-quarantine at four properties, and bioassays from collected soils indicate that CGMMV remained viable in at least two of the properties 12 months after plant hosts were removed from the ground. The infectivity of CGMMV from soil was also investigated in two trials with 140 watermelon seeds and 70 watermelon plants sown into CGMMV infested soils with or without the application of the disinfectants Virkon^TM^ (2%) and Bleach (1%). Watermelons grown in soil, not treated with the Virkon^TM^ or Bleach, showed CGMMV infection rates of 4% and 2.5% respectively. When Virkon^TM^ or Bleach was applied, no positive CGMMV detections were observed in the watermelons. This research highlights the importance of proper management of infested properties and the need for on-farm biosecurity to manage CGMMV.

## 1. Introduction

*Cucumber green mottle mosaic virus* (CGMMV) is a *Tobamovirus* of the family *Virgaviridae* first described in 1935; it has now been identified on almost every continent [1,2]. CGMMV is an economically important seed-borne virus of cucurbit crops and Asian cucurbit vegetables throughout the world and was recently identified in Australia [3]. Transmission of CGMMV, like many *Tobamoviruses*, occurs via two main routes, seed transmission and mechanical transmission [4,5,6,7,8,9].

The first detection of CGMMV in Australia occurred in September 2014 in the town of Katherine, Northern Territory (NT), 300 km south of Darwin. In the subsequent months, CGMMV was isolated on 26 properties spread over Katherine, Ti-Tree (1200 km south of Darwin) and Darwin and its rural areas. The detections in the NT ranged from watermelons, cucumbers, pumpkins, squash and Asian cucurbits to cucurbitaceous weeds. The total production values in the NT for melons in the 2016/2017 financial year was estimated to be $50.3 million, while that of vegetables (Asian and other vegetables) was estimated to be $43.9 million [10]. 

Since the initial detection in the NT in 2014, CGMMV has been found in Western Australia, isolated areas in Queensland [2,11], South Australia (http://www.melonsaustralia.org.au/ (accessed on 27 May 2021)) and New South Wales. Following the initial detections in the NT, commercial cucurbit properties found to be positive for the virus were put under quarantine for a period of two years. During this quarantine period no host plants were permitted to be grown on the properties and potential weed hosts were eradicated. Quarantine measures for CGMMV in the NT were lifted in February 2016.

The detection and survival of *Tobamoviruses* in soil have been studied including *Tomato mosaic virus* [12,13] and *Tobacco mosaic virus* [14,15,16], while CGMMV has been found to remain viable in soil when stored at 4 °C for a minimum of 10 months [17]. Although CGMMV may survive in soils for a substantial amount of time, the likelihood of healthy seedlings becoming infected has been shown to be relatively low. Research has determined that of 330 bottle gourds that developed from seed in infested soil only 10% of those became infected with CGMMV [8]. 

The use of bleach and Virkon^TM^ as disinfectants for viruses has been well established, with sodium hypochlorite solution at concentrations of 5.25% or lower, and a 1–2% (*w/v*) Virkon^TM^ solution shown to reduce and, in some cases, eliminate all viable virus when in contact for as little as 10 s [8,18,19]. Disinfecting of CGMMV has largely involved chemical treatments such as sodium hypochlorite and trisodium phosphate and heat treatments of potentially infected seeds, which have been shown to reduce the incidence of disease [6,20].

The first aim of this research is to investigate the longevity of CGMMV in host-free soils and to better understand the rate of infection in infested soils, utilising field trials and bioassays using collected soil from infested properties. The second aim of this research is to investigate the use of disinfectants to remove viable CGMMV from the soil. The information collected from this study may help with the management of properties infested with CGMMV, helping to reduce further spread of the virus.

## 2. Results

### 2.1. Field and Screenhouse Trials

Plant material was collected from each of the four Infected properties (IPs) eight weeks post planting and tested for the presence or absence of CGMMV (Table 1).

Of the four properties tested, only IP2 returned a positive result, with one bulk sample returning a positive result for CGMMV. The bioassays conducted on the soil collected from the four IPs revealed that at the initial sampling point of 12 months post quarantine, IP1 (one of eight bulk sample) and IP3 (two of eight bulk samples) were positive for CGMMV. Bioassays conducted on the soils collected at 15 months post quarantine revealed all four IPs tested positive to the virus with IP1 having four of eight bulk samples return a positive result, IP2 and IP3 having five of eight bulk samples test positive and IP4 having three of eight bulk samples return a positive result. Bioassays conducted on the two remaining IPs revealed that only IP4 (one of eight bulk samples) was positive for CGMMV (Figure 1).

#### 2.1.1. Infectivity of CGMMV in Soil

In the initial trial, 14 watermelon bulk samples (five pots per bulk sample) were tested for the presence or absence of CGMMV. Three of 14 bulked leaf samples (equating to 15/70 pots) tested positive for the presence of CGMMV using conventional PCR. To determine how many of the 15 pots tested positive (samples 5 and 13), each pot was sampled individually, with only 3/15 pots testing positive for CGMMV (Table 2).

*Nicotiana benthamiana* plants bulked into four samples (10 plants per sample) were tested for the presence or absence of CGMMV, with two of the four samples returning a positive result. To further analyse this, 19/20 (one sample did not survive to this point) *N. benthamiana* pots that made up the bulk samples were sampled individually. Testing of the individual pots revealed that 9/19 or 47% of *N. benthamiana* plants tested positive for CGMMV (Table 3). Overall, 9/79 individual pots, or 11.39%, returned a positive result.

#### 2.1.2. Virus and Soil Disinfection

*N. benthamiana* and watermelon plants inoculated with CGMMV that had been incubated in either 2% Virkon^TM^ or 1% Bleach at 30 s, 1 min or 5 min were tested for the presence of CGMMV. There was no detection of CGMMV through conventional and RT-qPCR for watermelon inoculated with either treatment at the three incubation times (Table 4).

*N. benthamiana* inoculated with CGMMV and incubated in 2% Virkon^TM^ did not reveal any positive results in RT-qPCR or RT-PCR at either incubation time. However, *N. benthamiana* plants inoculated with CGMMV and incubated in 1% bleach at all three time points initially revealed a positive result in one of three RT-PCRs (Movement Protein, Coat Protein and RNA Helicase Subunit), but not in RT-qPCR. This false positive was later sequenced to be host material.

To test the effectiveness of soil disinfection using bleach and Virkon^TM^, two bulk samples of watermelon and *N. benthamiana* were taken from each treatment eight weeks post planting. The results from the watermelon and *N. benthamiana* plants sown into CGMMV infested soil and treated with either 1% bleach or 2% Virkon^TM^ revealed that each of the bulk samples were positive for CGMMV.

Two weeks after the initial testing, 10 watermelon and 10 *N. benthamiana* plants were tested individually. The two sets of 10 watermelons individually sampled from the 1% bleach and 2% Virkon^TM^ treated soil in individual pots tested negative for CGMMV. For the two sets of 10 *N. benthamianas* individually sampled, seven of 10 *N. benthamiana* plants from the 2% Virkon^TM^ soil disinfection trial tested positive (Figure 2) and were confirmed as CGMMV using Sanger Sequencing, while all 10 *N. benthamiana* plants in the 1% bleach soil disinfection trial were CGMMV negative.

## 3. Discussion

### 3.1. Survival of CGMMV in Soil

*Cucumber green mottle mosaic virus* has the potential to decimate commercial cucurbit and melon farms, with proper management of the virus required to limit any potential spread. The *Tobamovirus* itself is extremely hardy and can survive in harsh conditions. In this study, watermelon plants in three of the four IPs tested positive for CGMMV in either the field trial or bioassays, which had initially been left bare for 12 months. This indicates that to significantly reduce the chances of reinfection, infested soil should be left for at least 12 months before any host plants are planted into the ground and any known weed hosts should be carefully managed. In waterlogged soil, CGMMV was able to survive for up to 33 months in the roots of bottle gourd, however in moist well-aerated soil, CGMMV was only found to survive for 17 months [21]. It is difficult to determine for all four IPs if weed management was performed consistently to reduce the potential for suspected weed hosts, including wild melons, to harbour and maintain viral titre in soils. 

Research has shown that *Solanum bahamense* (Key West Nightshade) is not only susceptible to several *Tobamoviruses* including *Tobacco mosaic virus* and *Pepper mild mottle virus* but was also found to be systemically infected [22]. When looking at IP4, no infection was present until 15 months post-quarantine. This may indicate a reinfection of the virus from an external source, which could include infection from wild melons, bees, which have been shown to spread CGMMV, [23] or via birds or animals such as camels or cows, which were routinely seen walking across the field beds. Commercial cucurbit and melon growers could adopt management programs that hasten the reduction of CGMMV infectivity within their soils. This may include planting crops that are not hosts of CGMMV to allow for viral titre to reduce while providing an alternative income. Another option which may directly impact the survival of CGMMV is to rotate where the crops are grown. For example, leaving a field bare for 12–18 months before growing host plants again, and alternating growing years with a second field may reduce the overall titre of the virus and its infectivity in both fields over time.

### 3.2. Infectivity of CGMMV in Soil

Infested soils pose serious economic risks to growers, not only in the cost of lost produce, but also the management practices that are required to reduce the spread of the virus. In this study, the use of an experimental host (*Nicotiana benthamiana*) and crop host were used to demonstrate the complexity of CGMMV infectivity from infested soil. When healthy bottle gourd seedlings were sown in contaminated soil, 2% of seedlings became infected with CGMMV [24]. The use of bottle gourd as a natural CGMMV host follows the results found in the two trials undertaken in this research in which 4% and 2.5% of watermelon plants tested positive for CGMMV. Over three separate trials [25], 860 watermelon seedlings were sown into contaminated soil and 21 infections were observed, an infection rate of 2.4%, again highlighting the low level of initial infection from contaminated soil. 

The use of *N. benthamiana* as an experimental host reveals the difficulty in comparing the likelihood of an initial infection in a crop host with infection in an experimental host, as 47% of individually tested *N. benthamiana* plants were shown to be infected with CGMMV. This finding highlights a requirement to test the rate of infection between other crop hosts and potential weed hosts/reservoirs. Another consideration is the use of a disinfectant within the soil itself to reduce and potentially remove all viable CGMMV prior to planting susceptible cucurbit crops.

### 3.3. Disinfection of Infested Soil

Disinfectants and heat treatment are routinely used to disinfect potentially virus, fungal and bacterial infected seeds [26,27,28]; however, the use of some of these has been shown to be ineffective in removing all viable CGMMV [29]. The use of disinfectants in soil has not been widely studied. In this research the use of bleach (1%) and Virkon^TM^ (2%) was used to disinfect infested soil with watermelon and *N. benthamiana* seedlings tested for the viability of CGMMV post-treatment. Of the two treatments, bleach 1%, was the most effective in keeping watermelon and *N. benthamiana* plants free from CGMMV, while Virkon^TM^ was only able to keep watermelon free from CGMMV. 

The initial detections of CGMMV in both watermelon and *N. benthamiana* from soil treated with 1% bleach may have picked up nonviable or low titre virus at the time of detection. The inability to detect the virus two weeks following the initial detection may indicate the virus’ inability to replicate and move through the hosts. This has been observed in manually infected *Datura stramonium*, which appears to only cause localised infections [30]. The observation of symptomology associated with CGMMV in *N. benthamiana* (Figure 2) in infested soil, treated with 2% Virkon^TM^, may indicate that the starting concentration requires further evaluation and testing to remove all viable virus. There were no such observations detected in any of the watermelon or *N. benthamiana* plants in infested soil that were treated with 1% bleach at any stage of the trial.

The difficulty in assessing the effectiveness of disinfectants in soil is highlighted by the low percentage of host plants that became infected when grown in untreated infested soils. Although the number of *N. benthamiana* plants that tested positive to CGMMV was extremely low compared to those grown in untreated infested soils, the fact that a positive detection was identified is a cause for concern when developing a strategy to combat the virus. Factors such as water content, temperature and soil composition along with weed hosts may play a role in the ability of CGMMV to survive in soils and requires further research. To understand these relationships better and for growers to better manage any future outbreaks, more research is required.

## 4. Materials and Methods

### 4.1. Infested Property Selection and Soil Sampling

Four property managers whose farms were under quarantine were contacted to take part in a trial to determine if CGMMV was still present and viable in differing NT soils. One property in Darwin, two properties in Katherine and one property between Ti-Tree and Alice Springs gave permission for us to take soil samples and conduct a field trial on their properties (Figure 3).

The four properties presented an opportunity to investigate the longevity of CGMMV in a range of soil types and varying conditions. Samples were taken at 12, 15 and 18 months post-quarantine. 

In Darwin and its rural areas, Kandosols are the most common soil type, while in Katherine and Ti-Tree there is a mixture of Kandosols and Tensols (www.irm.nt.gov.au (accessed on 1 February 2019)). Temperature is an important factor for the breakdown of host material; the wet season of 2015/2016 recorded an average minimum and maximum temperature in Darwin, Katherine and Ti-Tree as 25.6 °C and 33.6 °C; 24.2 °C and 36.2 °C; and 20.7 °C and 36.5 °C, respectively. The average minimum and maximum temperature in the dry season of 2016 in Darwin, Katherine and Ti-Tree were recorded as 22.8 °C and 32.9 °C; 19.1 °C and 34.4 °C; and 10.4 °C and 26.2 °C, respectively (www.bom.gov.au (accessed on 1 February 2019)).

Every positive detection for CGMMV had accompanying GPS coordinates in all four locations, this provided a focal point on each of the properties. The field beds were kindly prepared by the growers prior to any work being conducted on the properties. Three of the four properties had six rows, while one property had five field rows. The area of focus in each of the properties was a 12 × 12-m plot, with the GPS positive in the centre (Figure 4).

Within each plot, eighty soil samples were taken at a spacing of 75 cm in each of the beds prior to the planting of host plants, with approximately 1 kg of soil collected per soil sample to a depth of 50 cm, which was then stored in individual 1000 mL plastic containers. The containers were wiped with 2% Virkon^TM^ S (Livingstone International, Mascot, NSW, Australia) and sealed in larger containers, before being transported to the laboratory at Berrimah Farm Science Precinct (hereafter referred to as Berrimah Farm), Darwin, NT, Australia.

### 4.2. Plant Growth and Field and Screenhouse Trials

Four-week-old *Citrullus lanatus* (watermelon), *Cucumis sativus* (cucumber) and *Cucurbita pepo* (pumpkin/squash) plants were supplied by a commercial nursery (PlantSmith Nursery, NT) for all field and screenhouse trials relating to soil bioassays from the four properties. All plants were tested for CGMMV prior to any research being conducted to ensure that they were CGMMV negative.

Host plants chosen for the field sites represented what each property commonly grew at the time of CGMMV detection. Field beds had either underground or t-tape irrigation, fertilizer in soil or added through the water and mulch film placed over the top to reduce weed and grass growth. Insect netting, with holes small enough (2 mm × 2 mm) to prevent the introduction of bees, was erected over each of the field plots prior to planting of cucurbits to avoid any further spread of the disease and unnecessary disturbances to the plot. The netting was left in place for the entirety of the trial. Eighty plants were removed from punnets, their root systems roughly handled to cause slight amounts of damage to increase the likelihood of infection and planted ~75 cm apart at each of the properties. The plants were left to grow for eight weeks prior to sampling, removal and disposal of plant and other material. Eight bulk samples were collected at each property, with one leaf per plant collected and bulked to a maximum of ten plants. Leaf material was then tested for the presence or absence of CGMMV (see Isolation of viral RNA, PCR primers and conditions below).

Field soil collected from each of the four properties was stored at 4 °C to prevent viral degradation. The screenhouse utilised at Berrimah Farm was outfitted with automatic watering, with 80 drippers installed per bench, and five benches in total (Figure 5). 

In 100 mm pots, a small amount of potting mix (~2 cm) produced at Berrimah Farm, Coir chip/coir fines/fine pine bark (50%/35%/15%), was added to the base of the pots, with a single soil sample making up the rest of the volume. Each of the 80 soil samples from the four properties was added to individual pots and a dripper placed into the soil. A *C. lanatus* (cucumber), *C. sativus* (watermelon) or *C. pepo* (pumpkin/squash) plant was then planted into the soil, using the same root damaging technique as the field trial. Eighty cucurbit plants were placed into pots of the same size with potting mix and manually inoculated with CGMMV as a positive control and placed on the fifth bench with automatic watering, while another 80 cucurbit plants used as negative controls were potted and placed on a bench and hand watered twice daily. Plants were left to grow for eight weeks prior to sampling and storage, disposal of material and disinfection of the screenhouse.

### 4.3. CGMMV Inoculum Preparation and Disinfection of CGMMV

Dried *Nicotiana benthamiana* material (0.25 g) infected with CGMMV was ground in a 12 cm × 15 cm Bioreba^®^ (ThermoFisher Pty Ltd., Waltham, MA, USA) extraction bag with 5 mL of 0.01 M Potassium phosphate buffer (pH 7), with the resulting liquid transferred to a 15 mL centrifuge tube. A total of 10 mL of inoculum was added to the tube potassium phosphate buffer. This liquid was then used in all positive controls and re-made fresh when required.

To confirm the effectiveness of Virkon^TM^ and Bleach as disinfectants of CGMMV, inoculum was incubated in either 2% Virkon^TM^ (*w*/*v*) or 1% Bleach (*v*/*v*) at three-time points, 30 s, 1 min and 5 min. At the completion of the desired time point five watermelon or five *N. benthamiana* plants were inoculated by rubbing the inoculum into the lower leaves with the addition of silicon carbide to create wound openings on the surface of the leaf to allow any viable CGMMV to infect them. The plants were left for a period of 8 weeks prior to being tested for the presence of CGMMV.

### 4.4. Soil Infectivity and Disinfection

Watermelon and *N. benthamiana* seedlings for soil disinfection and infectivity trials were grown at Berrimah Farm and were approximately four weeks old at the time of each trial.

An initial trial for soil infectivity was undertaken with 110 pots containing soil infested with CGMMV. To each of the first 70 pots, two watermelon seeds were sown and left for a period of 12 weeks before being sampled, with five pots (one leaf per 10 plants) making up a single bulk sample. The remaining 40 pots each contained a single *N. benthamiana* plant and were left to grow for the same period. The *N. benthamiana* plants were then sampled with 10 pots per sample, equalling four bulk samples. After initial testing, 20 pots were individually tested for the presence or absence of CGMMV.

Soil from the initial infectivity trial and positive CGMMV material was mixed with a commercial potting mix (Bunnings Warehouse) in a 1:2 ratio. The soil was then divided into two different sized pots, 30 (100 mm) pots for soil disinfection and 40 (200 mm) pots for a second soil infectivity trial, with 10 mL of CGMMV extracted sap added to all pots. 

For the disinfection trial, the 100 mm pots were divided into three groups of ten; Positive control, 2% Virkon^TM^ and 1% bleach, a further ten (100 mm) pots with clean soil (Berrimah Farm mix) were also added as the negative control. To each of the test pots was added either 100 mL of 2% Virkon^TM^ or 1% bleach to a total of ten pots each, the disinfectants used were left to drain through the soil and out of the pots. To the positive and negative controls was added 100 mL of water, and after 24h all pots had two washes with 100 mL of water. To each of the 40 (200 mm) pots and 20 (100 mm) test pots were added two watermelon seedlings and one *N. benthamiana* seedling, while to the positive and negative control pots one watermelon seedling was added to each. All plants were left to grow for a period of 12 weeks before sampling. Bulk samples consisted of one leaf per plant with a maximum of 10 plants per sample for all treatments and controls.

### 4.5. Isolation of Viral RNA, PCR Primers and Conditions

Each of the bulk samples collected were roughly chopped and a subsample of the material placed into 12 cm × 15 cm Bioreba^®^ (ThermoFisher Pty Ltd., Waltham, MA, USA) extraction bags. Total RNA was then extracted using the Isolate II Plant RNA Kit (Bioline Pty Ltd., Taunton, MA, USA) according to the manufacturer’s instructions, with the following modification, 700 µL of RLY Buffer and 7 µL of β-mercaptoethanol were added to the extraction bag and a pestle was lightly used to grind the material before applying the liquid to the first column.

Conventional RT-PCRs were performed using a Veriti Thermal Cycler (ThermoFisher Scientific, Waltham, MA, USA), utilising Superscript III One Step with Platinum Taq (ThermoFisher Pty Ltd., Waltham, MA, USA), following the manufacturers guidelines. The following primer pairs were used for detection of CGMMV: Coat Protein (CP) (496 bp) [30] Forward primer, 5′-GATGGCTTACAATCCGATCAC-3′ and Reverse primer, 5′-CCCTCGAAACTAAGCTTTCG-3′; Movement Protein (MP) (809 bp) [31] Forward primer, 5′-TAAGTTTGCTAGGTGTGATC-3′ and Reverse primer, 5′-ACATAGATGTCTCTAAGTAAG-3′; CGMMV RNA helicase subunit (1053 bp) [32] Forward primer, 5′-ATGGCAAACATTAATGAACAAAT-3′ and Reverse primer, 5′-AACCACACAGAAAACGTGGC-3′. 

The cycling conditions for the detection of CGMMV using Superscript III were as follows: one cycle at 48 °C for 45 min and one cycle at 94 °C for 2 min, followed by 40 cycles of 94 °C for 40 s, annealing (MP 50 °C, CP 52 °C and CGMMV variable region 57 °C) for 40 s and extension at 72 °C for 40 s, a final extension was performed at 72 °C for 5 min, with PCR reactions held at 4 °C upon completion. Conventional PCR products were visualised on an agarose gel, amplified PCR products were cleaned up with an Isolate II PCR and Gel Kit (Bioline Pty Ltd., Taunton, MA, USA). The purified PCR products were sequenced in both directions at the Australian Genome Research Facility Ltd. (Brisbane, Australia www.agrf.org.au (accessed on 13 August 2018)) and analysis undertaken using Geneious^®^ (Version 8, Biomatters Ltd., Auckland, New Zealand) and compared to submitted sequences on BLAST (https://blast.ncbi.nlm.nih.gov (accessed on 17 August 2018)).

RT-qPCR was performed on a Rotor-Gene 6000 (Qiagen, Hilden, Germany) with SensiFAST^TM^ SYBR^®^ No-ROX One-Step Kit (Bioline Pty Ltd., Taunton, MA, USA) following the manufacturers conditions, using the following primer pair and probe [33]. Forward primer, 5′-GTGGTTTCTGGTGTATGGAACGTA-3′, Reverse primer, 5′-CGGGAGCTGAAAATTTGCATATAGT-3′ and probe (RZ_CGMMVmp-03), 5′-[FAM]-CACCCCTACAGGATTC-[NFQ-MGB]-3′. The following cycling conditions were used: an initial cycle at 45 °C for 10 min, followed by one cycle at 95 °C for 2 min and finally 35 cycles of 95 °C for 5 s and 60 °C for 20 s. Results of RT-qPCR were analysed using the supplied software. The RT-qPCR interpretation of results was as follows: Ct value < 30 was CGMMV positive, Ct values 30–34 were CGMMV suspects and Ct values > 34 were CGMMV negative. The threshold was set at the beginning of the exponential phase, to avoid background noise from the negative control and no template control.

For all plant samples tested for the presence and absence of CGMMV, the internal plant control primers AtropaNad2 1a (5′-GGACTCCTGACGTATACGAAGGATC-3′) and AtropaNad2 2b (5′-AGCAATGAGATTCCCCAATATCAT-3′) [34] were used to confirm successful extraction of host plant RNA, using MyTaq (Bioline, Pty, Ltd. Taunton, MA, USA), following the manufacturers guidelines with the following conditions: one cycle at 48 °C for 30 min, one cycle at 94 °C for 2 min, followed by 40 cycles of 94 °C for 1 min, 55 °C for 40 s and 68 °C for 40 s, followed by one cycle at 60 °C for 5 min, before the reaction was held at 4 °C.

## Figures and Tables

**Figure 1 plants-11-00883-f001:**
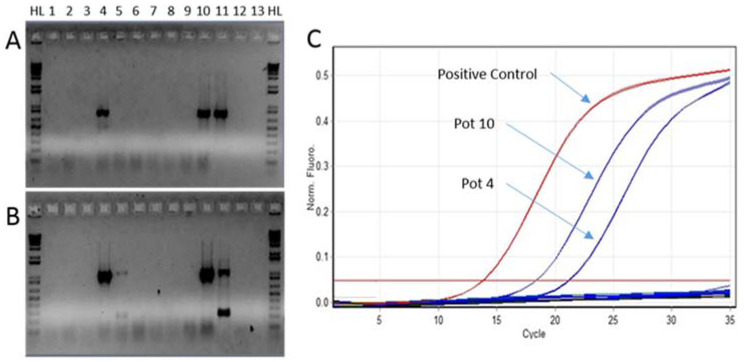
Conventional RT-PCR of the CGMMV coat protein (**A**), movement protein (**B**) and RT-qPCR (**C**) of positive CGMMV detections in soil longevity (18 month) screenhouse trials of IP4 (Lane 1–10 = pots 1–10; Lane 11 = positive control; Lane 12 and 13 = negative controls).

**Figure 2 plants-11-00883-f002:**
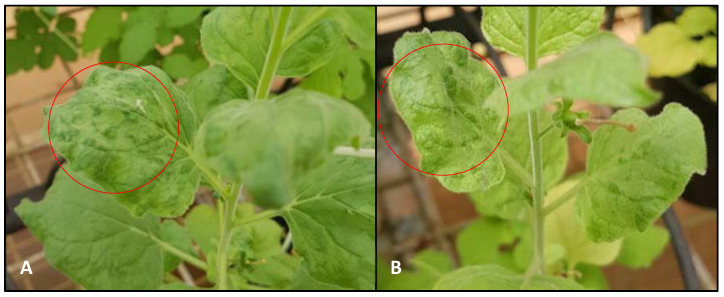
Symptomology (Red circles) of CGMMV on *N. benthamiana* planted into infested soil treated with (**A**) 2% Virkon^TM^ and (**B**) untreated soil.

**Figure 3 plants-11-00883-f003:**
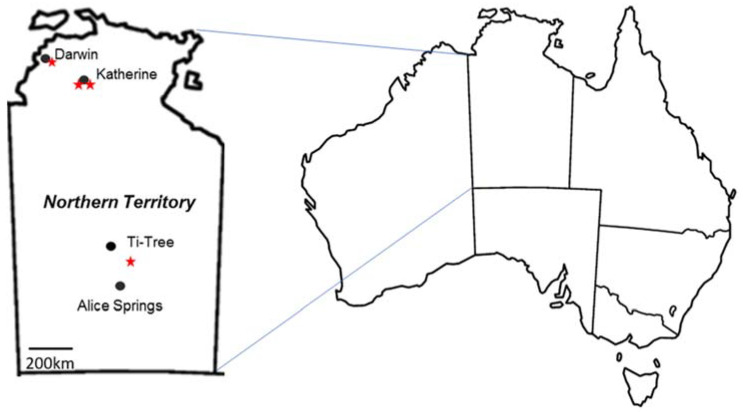
The location (red star) of the four filed trial sites spread out across the Northern Territory.

**Figure 4 plants-11-00883-f004:**
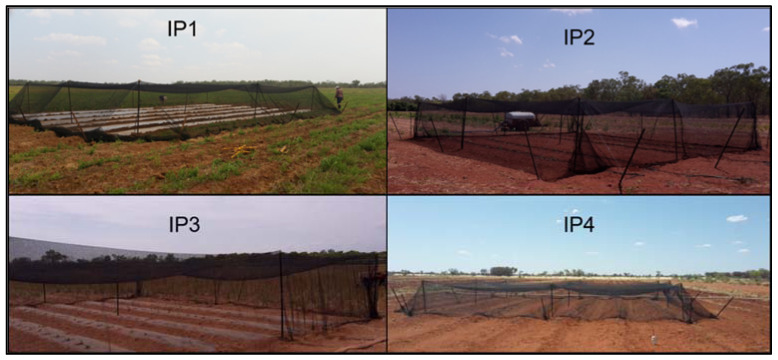
Completed field bed sites with insect proof netting at each of the four field site locations (IPs).

**Figure 5 plants-11-00883-f005:**
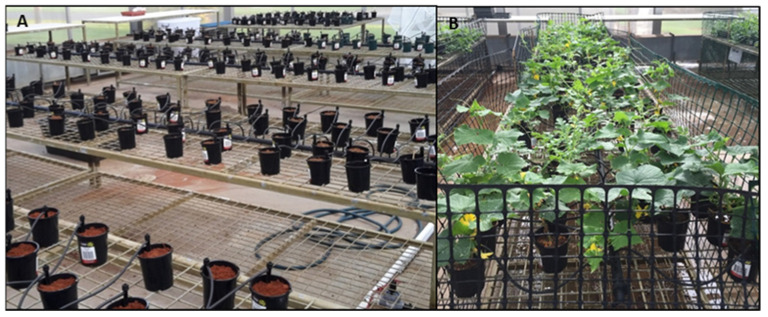
Setup of the soil longevity trial conducted in an insect proof screenhouse at Berrimah Farm Research Station. (**A**) Pre-planting; (**B**) Post Planting.

**Table 1 plants-11-00883-t001:** In-field and pot trial (Bioassay) results of the four IPs for detection of CGMMV at 12, 15 and 18 months-post quarantine.

Location	12 Month Field Trial	12 Month Bioassay	15 Month Bioassay	18 Month Bioassay
IP1	0/8 ^1^	1/8	4/8	NA ^2^
IP2	1/8	0/8	5/8	NA
IP3	0/8	2/8	5/8	0/8
IP4	0/8	0/8	3/8	1/8

^1^ Number of positive samples recorded from eight bulk samples. ^2^ NA = Not applicable, no samples were taken at this time-point.

**Table 2 plants-11-00883-t002:** Bulk sampling of 70 CGMMV infested pots containing watermelon plants and tested for the presence of CGMMV.

Sample	Positive/Total Pots	% Infected
1	0/5 ^1^	0 ^2^
2	0/5	0
3	0/5	0
5	1/5	20
6	0/5	0
7	0/5	0
8	0/5	0
9	0/5	0
10	0/5	0
11	0/5	0
12	0/5	0
13	2/5	40
14	0/5	0
Total	3/70	4

^1^ Number of positive pots detected from individually tested pots. ^2^ Percent of infection based upon positive pots against total pots.

**Table 3 plants-11-00883-t003:** *N. benthamiana* plants sown into CGMMV infested soil and tested for the presence of CGMMV.

Sample (Pot #)	Positive/Negative
1 (Pot 11)	− ^1^
2 (Pot 12)	+ *^,2^
3 (Pot 13)	−
4 (Pot 14)	−
5 (Pot 15)	−
6 (Pot 16)	−
7 (Pot 17)	−
8 (Pot 18)	−
9 (Pot 19)	+
10 (Pot 20)	−
11 (Pot 31)	+
12 (Pot 32)	+ *
13 (Pot 33)	−
14 (Pot 34)	+ *
15 (Pot 35)	+ *
16 (Pot 36)	+ *
17 (Pot 38)	+ *
18 (Pot 39)	−
19 (Pot 40)	+

^1^ −/+ Represents a negative or positive result. ^2,^* Represents confirmation of CGMMV through Sanger Sequencing.

**Table 4 plants-11-00883-t004:** *N. benthamiana* inoculated with CGMMV mixed with 2% Virkon^TM^ or 1% Bleach at three separate contact times.

Time Point	2% Virkon^TM^		1% Bleach		Negative	Positive
	Watermelon	*N. benthamiana*	Watermelon	*N. benthamiana* *^,2^	Watermelon	Watermelon
30 s	− ^1^	−	−	−	−	+
1 min	−	−	−	−	−	+
5 min	−	−	−	−	−	+

^1^ −/+ Represents a negative or positive result. ^2,^* False positives were detected at all timepoints for the Coat Protein (RT-PCR), which were confirmed through Sanger Sequencing.

## Data Availability

Not applicable.

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
