# Peer review of "Investigating the Longevity and Infectivity of *Cucumber green mottle mosaic virus* in Soils of the Northern Territory, Australia"

_plants, 2022, doi:10.3390/plants11070883_

Round 1
Reviewer 1 Report
Investigating the longevity and infectivity of Cucumber green mottle mosaic virus in the Northern Territory, Australia
This ms reports on studies to determine the longevity of Cucumber green mottle mosaic virus (CGMMV) in soil, and the effectiveness of Virkon and bleach drenches to eliminate infectivity of CGMMV in soil. This copy provides better information as to how the trials were conducted and methodology used in the trials. I would suggest that soil needs to be included in the title. I recommend acceptance after consideration of the reviewers’ comments.
Pg 5, line 140: Do you mean RT-PCR by the term “conventional tests”?
Pg 8, line 294: What was the size exclusion of the insect proof netting, and it is unclear if this was allowed to remain in place after the test plants were planted.
Author Response
The title has been altered to include Soils.
Pg 5, line 140: Do you mean RT-PCR by the term “conventional tests”?
This has been edited to now just say RT-PCR, to avoid confusion.
Pg 8, line 294: What was the size exclusion of the insect proof netting, and it is unclear if this was allowed to remain in place after the test plants were planted.
The size of the netting has now been included and updated wording to confirm the netting was in place for the entire trial.
Reviewer 2 Report
The manuscript of Lovelock et al investigates in a comprehensive manner the longevity and infectivity of Cucumber green mottle mosaic virus in infested soil as well as the disinfecting activity of Virkon and bleach. The subject has been investigated before, together with other tobamoviruses (like ToBRFV) however it is always welcome to accumulate knowledge from different experimental, edaphological and climate conditions. In general the work is well executed and presented.
Two minor issues:
-Please do not refer to N. benthamiana as tobacco plants as this is used normally for N. tabacum, a non-host of the virus.
- On what base the RT-qPCR was used with thresholds (lines 403-404) ?
Author Response
Please do not refer to N. benthamiana as tobacco plants as this is used normally for N. tabacum, a non-host of the virus.
This has been changed to N. benthamiana instead of tobacco where appropriate.
On what base the RT-qPCR was used with thresholds (lines 403-404)?
The threshold was set at the beginning of the exponential phase, to avoid background noise from the negative control and no template control. – This has been added into the manuscript.
This manuscript is a resubmission of an earlier submission. The following is a list of the peer review reports and author responses from that submission.